# Chirality and energy transfer amplified circularly polarized luminescence in composite nanohelix

Dong Yang[1,2,3], Pengfei Duan[2], Li Zhang[1] & Minghua Liu[1,2,3,4]

Transfer of both chirality and energy information plays an important role in biological systems. Here we show a chiral donor π-gelator and assembled it with an achiral π-acceptor to see how chirality and energy can be transferred in a composite donor–acceptor system. It is found that the individual chiral gelator can self-assemble into nanohelix. In the presence of the achiral acceptor, the self-assembly can also proceed and lead to the formation of the composite nanohelix. In the composite nanohelix, an energy transfer is realized. Interestingly, in the composite nanohelix, the achiral acceptor can both capture the supramolecular chirality and collect the circularly polarized energy from the chiral donor, showing both supramolecular chirality and energy transfer amplified circularly polarized luminescence (ETACPL).

[1] Beijing National Laboratory for Molecular Science, CAS Key Laboratory of Colloid, Interface and Chemical Thermodynamics, CAS Research/Education Center for Excellence in Molecular Sciences, Institute of Chemistry, Chinese Academy of Sciences, No. 2 ZhongGuanCun BeiYiJie, Beijing 100190, China. [2] CAS Center for Excellence in Nanoscience, Division of Nanophotonic, CAS Key Laboratory of Nanosystem and Hierarchical Fabrication, National Center for Nanoscience and Technology (NCNST), No. 11 ZhongGuanCun BeiYiTiao, Beijing 100190, China. [3] University of Chinese Academy of Sciences, Beijing 100049, China. [4] Collaborative Innovation Centre of Chemical Science and Engineering, Tianjin 300072, China. Correspondence and requests for materials should be addressed to P.D. (email: duanpf@nanoctr.cn) or to M.L. (email: liumh@iccas.ac.cn).

As one of the most important structural characteristics and the existing form of biological information, chirality is ubiquitous in nature and organisms, which is best exemplified by α-helix proteins, double-helical DNA and triple helix in collagen. In biological systems, transfer of both chirality and energy information at hierarchical assembly structures from molecular, macromolecular to nanoscale levels is crucial for the well implementation of their sophisticated functions involving recognition, replication and catalytic activity in life process[1–12]. In addition, transfer of information in living organisms is muti-channel, not only through chirality-based structural information, but also through ion-, electron- and light-based energy types[13–16]. Inspired by these sophisticated information communications in physiological systems, many researchers have tried to mimic the chiroptical material systems through a molecular to supramolecular level by self-assembly strategy[2,7,8,10,17–21]. Such process has been mimicked in various solution and liquid crystal systems[18,22]. However, both chirality transfer and circularly polarized luminescence (CPL) were generally separately reported, it remains unknown when the two processes were coupled. Self-assembly plays a significant role in the formation of many chiral biological structures as well as the functional chiral materials. Thus, control of the supramolecular chiroptical activity has rapidly evolved into one of the major research topic in supramolecular chemistry and nanotechnology[23–26]. In self-assembly system, not only one component, but also multi-component can be involved in self-assembly system, which usually generate amplified or new functions from these multi-component nanosystems. In addition, one of the greatest merits of the self-assembly system is that complex hierarchical chiral organization processes and chiral structures can be built rapidly with minimal synthetic efforts. Even achiral molecules can get chiroptical information from chiral donor at the supramolecular level through self-assembly route[27–30].

Herein, to investigate the multichannel chiral information transfer, we design a self-assembly system based on the chiral donor and achiral acceptor. With this design, we hope to solve two important questions. One is that if the molecular chirality of donor can be transferred to achiral acceptor to construct a donor–acceptor (D–A) nanosystem. Since the excited state of donor or acceptor can possibly have energy, thus, the other is that during such chirality transfer process, how energy transfer is related to the chirality? As expected, we find that the chiral gelator could form chiral nanohelix through gelation and show both supramolecular chirality and CPL. When an achiral acceptor, 9,10-bis(phenylethynyl)anthracene (BPEA), was mixed with the gelator molecules, a D–A composite nanohelix was formed. Interestingly, chirality transfer could be realized by just weak π–π interaction other than common hydrogen bonding[31], electrostatic interaction[32] or chain interdigitation[33,34]. More interestingly, in this composite nanohelix assemblies, the acceptor BPEA could capture the energy from the chiral gel through highly efficient energy transfer. Unexpectedly, during such energy transfer, not only the chirality of gelator was transferred to the D–A nanohelix, but also the circularly polarized fluorescence was significantly amplified. Such energy transfer amplified circularly polarized luminescence (ETACPL) provided a way to amplify the CPL. Although supramolecular chirality transferred from the chiral species to the achiral component and the production of CPL has been reported, the study presented herein highlights the ETACPL in the self-assembly nanoscale systems.

## Results

**Gel formation and self-assembled composite nanohelix.** The designed gelator (L-1 or D-1 for short) is shown in Fig. 1, which contains a donor chromophore, cyano-substituted stilbene (CNSB) conjugated with a universal gelator moiety $N,N'$-bis(dodecyl)-L(D)-amine-glutamic diamide. The detailed synthetic route was shown in Supplementary Fig. 1. It was found to show excellent gel ability in many kinds of organic solvents varying from polar to non-polar, and the water can strengthen the gel ability of L-1 or D-1. BPEA acceptor can be dissolved in the DMSO, but water as the poor solvent can motivate the BPEA to

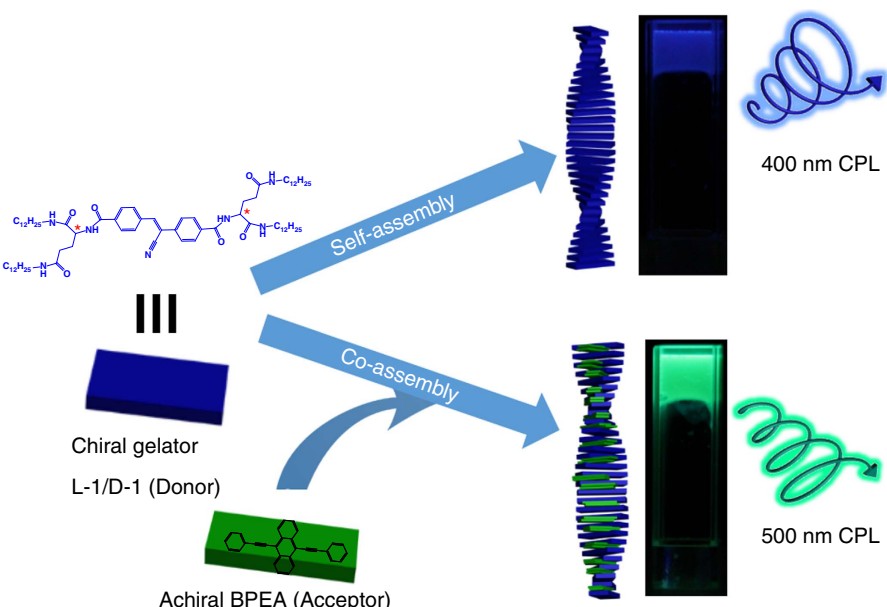

**Figure 1 | Chirality and energy transfer in the self-assembly system.** Chemical structures of L-1(D-1) and BPEA and schematic representation of chirality transfer and energy transfer between chiral gelation-amplified assemblies and achiral acceptor BPEA. (CPL = circularly polarized luminescence). Chiral gelator L-1(D-1) formed nanohelix and showed both supramolecular chirality and CPL. When an achiral acceptor, BPEA, was mixed with the gelator molecules, the BPEA acceptor was co-assembled into the nanohelix through just weak π–π interaction. In the combined nanohelix assemblies, BPEA could capture the energy from the chiral gel, thus the co-assemblies exhibited only the emission spectrum of BPEA.

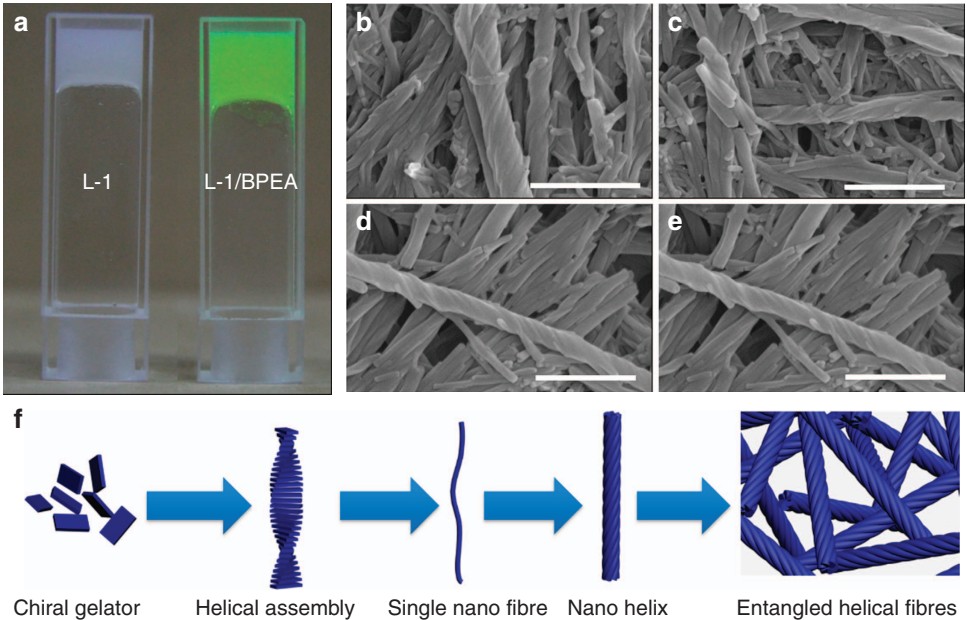

**Figure 2 | The self-assembled chiral structure at nanoscale for donor and D–A complex.** (**a**) The photo images for the gel of L-1 and L-1/BPEA; SEM images of xerogel made from (**b**) L-1, (**c**) D-1, (**d**) L-1/BPEA, (**e**) D-1/BPEA. (L-1 or D-1/BPEA = 5/1, molar ratio); (**f**) illustration of the possible self-assembly route for the gelation. Scale bars, 1 μm.

aggregate in DMSO. To drive the BPEA molecule to co-assemble with L-1 or D-1, water was added as the co-solvent to study the co-assembly behaviour. We studied the effect of water content with 1%, 5%, 10%, 15% ($H_2O/H_2O + DMSO$, v/v) on the supramolecular assemblies. It was found that the L-1 or D-1 molecule could not completely dissolve in mixed solvent when excess amount of water of 15% was employed even under heating. At the same time, BPEA acceptor will precipitate from the co-solvent system at 15% water amount. Thus, we selected 10% of $H_2O/DMSO$ as the mixed solvent.

To clarify the gel structures, we measured the morphology of the gels by scanning electron microscopy (SEM), as shown in Fig. 2. L-1 gelator formed right-handed nanohelix structures, while D-1 gelator with the opposite molecular chirality formed left-handed nanohelix structures (Fig. 2b,c). The SEM observation indicated that the molecular chirality of L-1 or D-1 gelator transferred to the supramolecular self-assemblies at the nanoscale during the gelation process and the supramolecular chirality is dominated by the chirality of the gelator. When BPEA acceptor was mixed with the gelator molecules, the assembled helical fibres exhibited same structures without obvious phase separation or other aggregation structures as shown in Fig. 2d,e. These results indirectly confirmed that, in the L-1(D-1)/BPEA co-gel system, BPEA could perfectly disperse into the co-gel system without phase separation or serious aggregation. We further performed stress-sweep rheological measurement to study the strength of the gel before and after the insertion of the acceptor BPEA. As shown in Supplementary Fig. 2, the storage modulus (G′) of the single-component D-1 gel proved to be ∼1.5 orders of magnitude larger than the loss modulus (G″), which is consistent with solid-like behaviour. While in the D-1/BPEA two-component gel, the storage modulus (G′) of the single-component D-1 gel proved to be ∼2.5 orders of magnitude larger than the loss modulus (G″). Obviously, the acceptor BPEA reinforced the magnitude of the storage modulus. Further, the presence of BPEA also resulted in increase of magnitude in the yield stress. These observations indicate that the insertion of the acceptor moiety leads to enhancement in the strength of the gel. From the SEM observation, one possible self-assembly route for the gelation

was as follows (Fig. 2f): Chiral molecules first formed helical assembly. Then several helical assembles twisted into larger fibres and several fibres further formed nanohelix. The nanohelix finally entangled with each other to gelatinize the solvent.

**Energy transfer in the nanohelix.** The CNSB is a well-known compound with aggregation-induced emission (AIE) property[35]. To endow the chiral gelator with AIE effect, we conjugated the gelator moiety N,N′-bis(dodecyl)-L(D)-amine-glutamic diamide with CNSB moiety to get L-1 and D-1 (refs 29,34). Actually, L-1 and D-1 showed gelation- or assembly-induced emission enhancement in various solvents. As shown in Supplementary Fig. 3a, the L-1 or D-1 molecule formed stable gel in $DMSO/H_2O$ (v/v = 9/1) at 25 °C and the system exhibited strong fluorescence emission. While the emission intensity decreased gradually as the temperature increased. The system changed to hot solution when the temperature arrived to 70 °C. In this temperature, the gelators existed in a molecularly dispersed state and the emission intensity fell close to zero. The temperature-dependent fluorescence spectra clearly indicated that the gel showed 100 times stronger emission at 445 nm than that of isolated single molecule in hot solution. On further mixing BPEA with the L- or D-1 in $DMSO/H_2O$ solvent, co-assembly occurred. Figure 3a presents normalized absorption and emission spectra of L-1 and BPEA in $DMSO/H_2O$ (v/v = 9/1) mixed solvent. L-1 gel exhibited fluorescence with a maximum at 445 nm and BPEA exhibited absorption from 380 to 500 nm, which matches the emission of L-1. Thus, the BPEA may be a proper energy acceptor from excited L-1. Furthermore, the emission of BPEA in $DMSO/H_2O$-mixed solvent exhibited similar emission (Supplementary Fig. 4b). By carefully characterizing the spectra of BPEA in various conditions, we found that both the absorption and emission showed almost the identical shape and position in solution, mixed-solvent solution and gel state at the same concentration (Supplementary Fig. 4). Considering that π–π stacking in aggregates always induced bathochromic shift of emission peak and quenched fluorescence emission related to the formation of excimer or exciplex in general π-conjugated

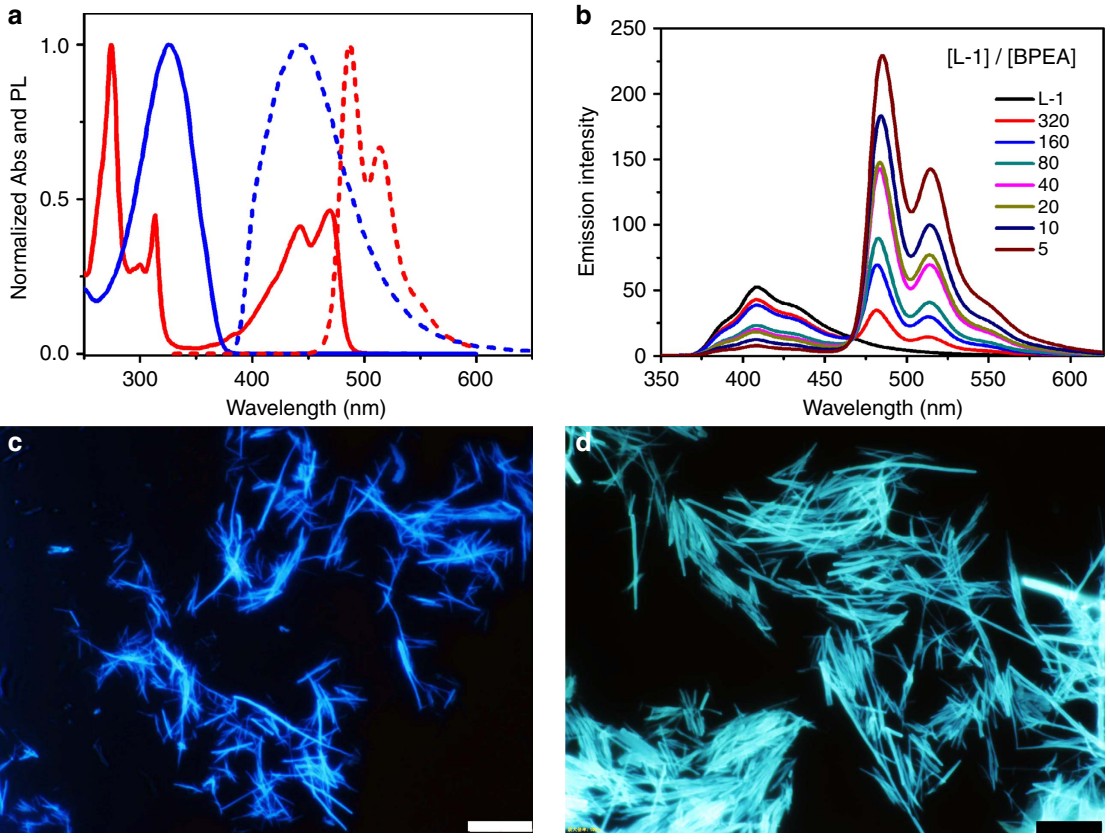

**Figure 3 | Energy transfer from donor to acceptor in the supramolecular gel.** (**a**) Normalized absorption (solid line) and emission (dashed line) spectra of the L-1 gel (blue line, [L-1] = 2 mM) and BPEA solution (red line; [BPEA] = 0.4 mM) in DMSO/$H_2O$ (v/v = 9/1)-mixed solvent. (**b**) Fluorescence spectral change in assembled L-1 (2 mM) induced by BPEA addition at different molar ratio: DMSO/$H_2O$ (v/v = 9/1), 25 °C, [BPEA] = 0–0.4 mM, $\lambda_{ex}$ = 320 nm. In the experiment, the concentration of L-1 was kept constant and the concentration of BPEA was varied. The digit (320, 160, 80, 40, 20, 10, 5) in the figure notes represents the molar ratio of L-1 to BPEA. Fluorescent images of (**c**) L-1 gel and (**d**) L-1/BPEA gel: DMSO/$H_2O$ (v/v = 9/1), [L-1] = [D-1] = 2 mM, [BPEA] = 0.4 mM, $\lambda_{ex}$ = 325–375 nm. Scale bars, 10 μm.

luminescent systems[36–39], L-1 with stable emission peak position is a proper energy donor in the self-assembly system. Keeping the same concentration of L-1 at 2 mM in DMSO/$H_2O$ (v/v = 9/1), by gradually adding BPEA to L-1 gel, the emission of L-1 decreased while the emission peak of BPEA increased with an isostilbic point at 467 nm (Fig. 3b). By increasing the ratio of BPEA, energy transfer could be more efficient. To further confirm the energy transfer in the co-gel system, we measured the fluorescent microscopy of the pure L-1 gel and L-1/BPEA co-gel in DMSO/$H_2O$-mixed solvent. As shown in Fig. 3c, L-1 showed deep-blue fluorescence while L-1/BPEA co-gel exhibited cyan-green emission (Fig. 3d). The emission colours are consistent with spectral measurements. More importantly, we can observe fibrous structures in both L-1 gel and L-1/BPEA co-gel without obvious phase separation. We have also tested the fluorescent microscopy and fluorescence spectrum of pure BPEA cast film (Supplementary Fig. 5). The aggregates of BPEA exhibited orange fluorescence while the fluorescence spectrum showed serious redshift with strong excimer emission band centred at 590 nm. These results indirectly confirmed that, in the L-1/BPEA co-gel system, BPEA could perfectly disperse into the co-gel system without phase separation or serious aggregation. For a better clarification of the efficient energy transfer, the relative fluorescence intensities of BPEA to L-1 were plotted against their molar ratio on excitation at 320 nm (Supplementary Fig. 6). However, no stable co-gel could be obtained when the ratio of L-1/BPEA reached to 1/1. This suggests that excess of BPEA may destroy the supramolecular assembly of L-1. Moreover, excess of

BPEA in co-gel showed aggregation-induced emission quenching (Supplementary Fig. 7). So we selected L-1/BPEA = 5/1 as the research object for the following discussion. The emission spectra of homo- and co-gels at 400 and 500 nm were very stable, which could be confirmed by time-dependent FL measurement. As shown in Supplementary Fig. 8, by naturally cooling down the hot solutions to room temperature, time-dependent fluorescence intensity of L-1 monitored at 410 nm and L-1/BPEA co-gel monitored at 485 nm showed a gradual increasing and finally reached a stable state. Energy transfer efficiency could be estimated by comparing the integrated area of L-1 emission spectra before and after adding BPEA. In the case of L-1/BPEA = 5/1, the efficiency was estimated with a relative high value of 86%. We can also compare the emission intensity of BPEA before and after dispersing into L-1 (Supplementary Fig. 4b). After collecting the energy from L-1, BPEA exhibited stronger emission than BPEA itself. It is around four times higher than the pure BPEA emission intensity. To further clarify the energy transfer process, we conducted the fluorescence lifetime measurement to study the possible mechanism. As shown in Supplementary Fig. 9a, in the absence of BPEA acceptor, the D-1 gel exhibited triple exponent decay with time constants of $\tau_1$ = 0.48 ns (71%), $\tau_2$ = 2.06 ns (18%) and $\tau_3$ = 10.01 ns (11%). The calculated average time of $\tau$ was about 1.15 ns. However, the emission decay of D-1 becomes faster in the presence of BPEA. When 20 mol% BPEA was added, D-1 exhibited a fast triple exponent decay with time constants of $\tau_1$ = 0.43 ns (71%), $\tau_2$ = 1.39 ns (22%), $\tau_3$ = 7.02 ns (7%) and the calculated average

time of $\tau$ was about 0.52 ns. The shortening of the averaged emission decay time of the D-1 donor in the presence of acceptor BPEA indicated that Förster mechanism may behave as the major mechanism for energy transfer[40–42]. We also illustrate Jablonski energy diagram including $S_0$, $S_1$ and $S_2$ states of the chiral donor and achiral acceptor in Supplementary Fig. 9b. The Jablonski energy diagram indicated that our system obeys Kasha's rule during the energy transfer process[43].

It should be noted that the highly efficient energy transfer process only occurred in the self-assembled nanohelix. That is, without the existence of nanohelix or the BPEA acceptor far away from the nanohelix formed by the L-1 or D-1 gelator donor, the effect would be low. As shown in Supplementary Fig. 10, the energy transfer process was still observed for the fluorescence spectra of L-1 and L-1/BPEA = 5/1 in dilute CHCl₃. The calculated efficiency was about 65%. In the dilute solution state of L-1/BPEA, the L-1 donor and BPEA acceptor were molecularly dispersed in the solvent. Thus, the L-1 and BPEA are far from each other and the relative orientation is rapidly changing. Therefore, the energy transfer process was restricted. However, in the self-assembled nanohelix system, the amplified interactions between L-1 donor and BPEA acceptor facilitated the process. There has been report that the energy transfer process occurred exclusively from nanostructured π-conjugated self-assemblies but not directly from the individual donor molecules[38]. In present example, the process can occur from both the self-assembled aggregates and individual molecule. The self-assembled nanohelix just facilitated the transfer process. Thus, a light harvesting supramolecular gel was successfully fabricated based on chiral a D–A nanosystem[44–46].

X-ray diffraction (XRD) measurements were performed to investigate the molecular packing of the L-1 and L-1/BPEA xerogel. In Supplementary Fig. 11a, the XRD profile of L-1 showed a series of peaks at 1.62°, 3.2°, 4.8° and so on. The corresponding distances were estimated to be 5.4, 2.7, 1.84 nm and so on according to the Bragg's equation. The $d$-spacing ratio is about 1:1/2:1/3, which is consistent with the characteristic of lamellar packing of the molecules[47]. As for L-1/BPEA, the XRD pattern was identical to the one of pure L-1, which confirmed that the addition of BPEA to L-1 did not destroy the well-ordered packing of L-1. Fourier transform-infrared (FTIR) spectroscopy was again used to gain insight into the gel formation and their driving forces. As shown in Supplementary Fig. 11b, the appearance of an N–H stretching vibration band at 3,296 cm⁻¹ for L-1 xerogel indicated the formation of the hydrogen bond[47]. The amide I band and II band appeared at 1,638 cm⁻¹ and 1,553 cm⁻¹, respectively, which indicates that both C=O and N–H are in the hydrogen-bonded form. The FTIR date confirmed the hydrogen bond interaction between the amide bonds of L-1 molecule is one of the driving forces for the gel formation. As for L-1/BPEA, the N–H stretching vibration band, amide I band and II band were identical to that of pure L-1, which further confirmed that the addition of BPEA to L-1 did not destroy the well-ordered packing of L-1.

**Supramolecular chirality and chirality transfer**. Since the L-1 and D-1 gelator have a chiral centre localized at the glutamic diamide, we measured their CD spectra to study the supramolecular chirality. As shown in Fig. 4a, the gels of L-1 and D-1 showed mirror image with a strong Cotton effect showing split peaks located at 331 nm, which is corresponding with the absorption spectrum. Considering that the CNSB core is an achiral moiety, the supramolecular chirality of L-1 and D-1 could be explained that the molecular chirality of glutamate transferred to the CNSB core assemblies[5,30]. When mixing L-1 or D-1 with

small amount of BPEA, co-gels could be achieved. Very interestingly, though BPEA is achiral, the co-gels showed the induced circular dichroism signals of BPEA, as shown in Fig. 4b. By carefully examining the CD signal, we found that, while the CD bands ascribed to the CNSB remained, there are two additional positive ICD signals for L-1/BPEA, whereas two negative CD signals for D-1/BPEA located at 445 and 470 nm, which are consistent with the absorption peaks of BPEA observed in gels. The induced CD signal of BPEA is strongly suggestive that the acceptor BPEA could capture the supramolecular chirality from the chiral nanohelix assemblies. Interestingly, the sign of the CD signals ascribed to the BPEA component is just opposite to that of L-1 or D-1. This means that the alignment of the acceptor is an opposite direction to the donor assemblies. We further gave a detailed discussion on the origin of bisignate CD band in the range of 420–500 nm. In Supplementary Fig. 12a, the CD spectra for L-1/BPEA or D-1/BPEA gels in DMSO/H₂O showed very complicated exciton couplet. This situation is partly due to that the $S_0$–$S_1$ transition absorption with two vibronic peaks for BPEA molecule is close together. The split-type CD signals for these two peaks have partial overlap. On the other hand, the strong CD signals for L-1 or D-1 molecules in the range of 200–420 nm will have interference in the CD signals of BPEA. In fact, the L-1/BPEA or D-1/BPEA formed semi-transparent gel in DMSO/H₂O, which will induce scattering effect in the CD measurement. This can be well illustrated in Fig. 4, the CD value of the baseline was high. Considering the scattering effect in the CD measurement for the gel in DMSO/H₂O, we are very careful to analyse these split CD signals. To gain insight into the state of BPEA (monomer or aggregate) in the gel, we gave a comparison of UV–vis spectra between D-1/BPEA gels in DMSO/H₂O and BPEA solution in CHCl₃ (Supplementary Fig. 12b). The UV–vis spectrum of BPEA solution in CHCl₃ (good solvent for BPEA) showed two monomer absorption peaks at 438 and 464 nm. While the L-1/BPEA or D-1/BPEA showed two absorption peaks at 445 and 470 nm, which are bathochromic shift compared with that in CHCl₃. This indicated that BPEA molecules mainly existed in the form of J-like aggregation in the gel formed by L-1 or D-1 in DMSO/H₂O. The CD spectrum for L-1/BPEA showed two positive bisignate CD bands in the range of 420–500 nm. The first positive bisignate CD band: 470 nm (positive), 459 nm (negative), 464 nm (the crossover); the second positive bisignate CD band: 445 nm (positive), 429 nm (negative), 438 nm (the crossover). It should be noted that the crossover in this case is just located at the monomer absorption peak (438 nm), while the positive peak corresponds to the maximum absorption peak at 470 nm. This clearly indicated the positive exciton couplet, which is just opposite to the L-1 exciton, suggesting that the chirality transfer in a multi to multi-mode. Moreover, a mirror CD can be observed for the D-1/BPEA system. On the other hand, the second bisignate CD band is not well resolved, mainly due to the interference from the strong CD signal of L-1 or D-1 in the range of 200–420 nm. It should be also noted that the peak seeming-like CD signal at about 490 nm arose from the scattering effect in the measurement process. The results strongly suggest that the supramolecular chirality rather than the chirality at the L-1 or D-1 is responsible for the ICD of BPEA.

Since the CD measurement of the gel system could contain some linear dichroism (LD) artifacts, we carefully studied the LD spectra and estimated the contribution of LD effect to the true CD intensity (Supplementary Figs 13–17). When rotating the sample for D-1 gel and D-1/BPEA gel about the optical axis in steps of 10°, the angle dependence of LD amplitude adopted cosine function and was positioned around the zero line. On the basis of the measurement, the contamination of CD by the LD artifact was evaluated to be 0.26% according to the semi-empirical

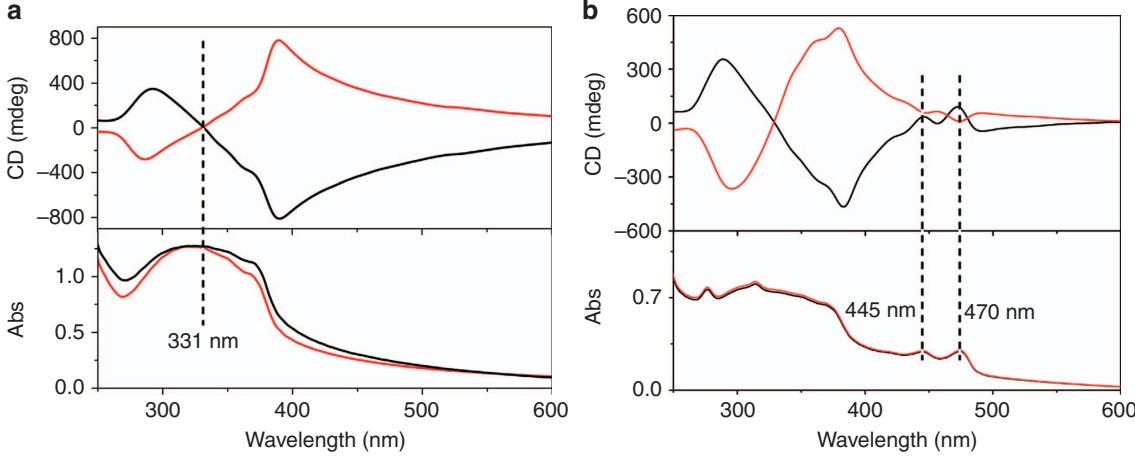

**Figure 4 | Chirality transfer from chiral donor to achiral acceptor in supramolecular level.** CD and UV–vis spectra for L-1 (black line) and D-1 (red line) gels in the absence (**a**) and presence (**b**) of BPEA. DMSO/$H_2O$ (v/v = 9/1), [L-1] = [D-1] = 2 mM, [BPEA] = 0.4 mM. It is clear that co-gels of L-1 or D-1 with BPEA showed ICD of BPEA located at 445 and 470 nm. A strong Cotton effect showing split peaks located at 331 nm for L-1 and D-1 gel. In L-1/BPEA co-gel, two positive ICD signals could be observed while two negative ICD signals were observed in D-1/BPEA co-gel.

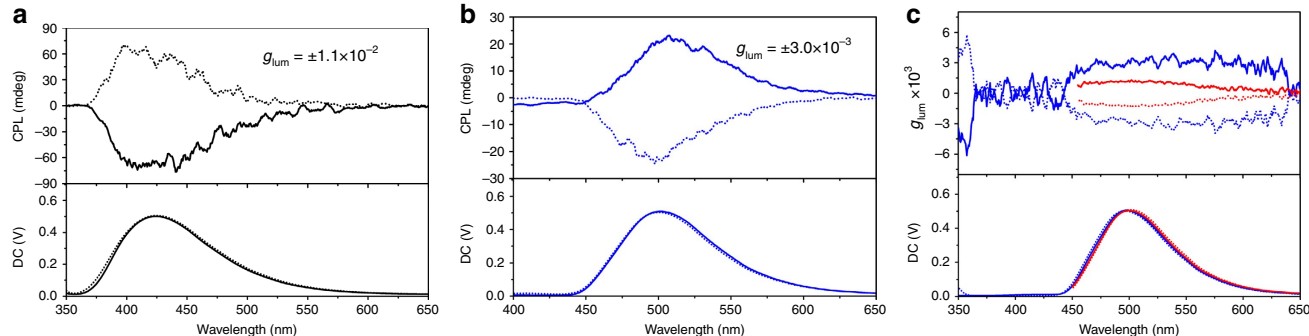

**Figure 5 | ETACPL.** (**a**) CPL spectra of L-1 (solid line) and D-1 (dash line) gels in the absence and (**b**) presence of BPEA excited at 320 nm; (**c**) CPL dissymmetry factor $g_{lum}$ versus wavelength of L-1/BPEA (solid line) and D-1/BPEA (dash line) co-gel excited at 320 nm (blue line) and 400 nm (red line). DMSO/$H_2O$ (v/v = 9/1), [L-1] = [D-1] = 2 mM, [BPEA] = 0.4 mM.

equation reported in literature[48–50]. The results show that the contamination of the CD spectra by LD artifacts was negligible. Considering that BPEA is an achiral dye without any active non-covalent site, the obtained CD signal could be assigned to the transfer through the weak π–π stacking between CNSB core and BPEA. To date, most of the reports about chirality transfer from the chiral centre or chiral assemblies to the achiral molecules focused on strong non-covalent interactions such as H-bond and electrostatic interaction while very few example involved in pure weak π–π interaction[51].

**CPL and amplification through energy transfer.** CPL is a unique property pertaining to the chiral system, which can be used to evaluate the excited-state supramolecular chirality of gels. Since L-1 and D-1 showed significant gelation- or assembly-induced fluorescence enhancement, we have further investigated the CPL response of these supramolecular gels. Amazingly, strong CPL signals with different handedness and emission maximum at 405 nm can be observed, as shown in Fig. 5a. For understanding the relationship between the ground-state supramolecular chirality and excited-state supramolecular chirality of gels, the correlation between the CD signs and CPL signs was studied. The results reveal that the sample L-1 with a negative Cotton effect display right-handed CPL, while the sample D-1 with a positive Cotton effect displays left-handed CPL.

The magnitude of CPL can be evaluated by the luminescence dissymmetry factor ($g_{lum}$), which is defined as $g_{lum} = 2 \times (I_L - I_R)/(I_L + I_R)$, where $I_L$ and $I_R$ refer to the intensity of left- and right-handed CPL, respectively[52]. The maximum $g_{lum}$ value ranges from $+2$ for an ideal left CPL to $-2$ for an ideal right CPL, while $g_{lum} = 0$ corresponds to no circular polarization of the luminescence. Experimentally, the CPL was measured using a JASCO CPL-200 spectrometer, and the value of $g_{lum}$ is defined as $g_{lum} = 2 \times$ [ellipticity/(32980/ln10)]/total fluorescence intensity at the CPL extremum. The calculated value of the dissymmetry factor ($|g_{lum}|$) of the CPL signal is about $1.1 \times 10^{-2}$ (Fig. 5a), which is a relatively large value compared with the ones reported in solution or in solid state[53–55]. More interestingly, after mixing with BPEA, the CPL peak of the co-gels shifted from 405 to 500 nm, as shown in Fig. 5b. The handedness of CPL for BPEA is consistent to the direction of induced CD of BPEA (Fig. 4b). The results reveal that the sample L-1/BPEA with two positive ICD signal displays left-handed CPL, while the sample D-1/BPEA with two negative CD signal displays a right-handed CPL (Fig. 5b).

It should be noted that after mixing with BPEA, the CPL signal located in the fluorescence emission zone of BPEA exhibited positive peak for L-1/BPEA and negative peak for D-1/BPEA accompanied by the disappearance of CPL signal of L-1 or D-1. This result is presumably due to the circularly polarized energy from the L-1 or D-1 chiral assembly collected by BPEA. To confirm the energy transferred CPL, we further measured the

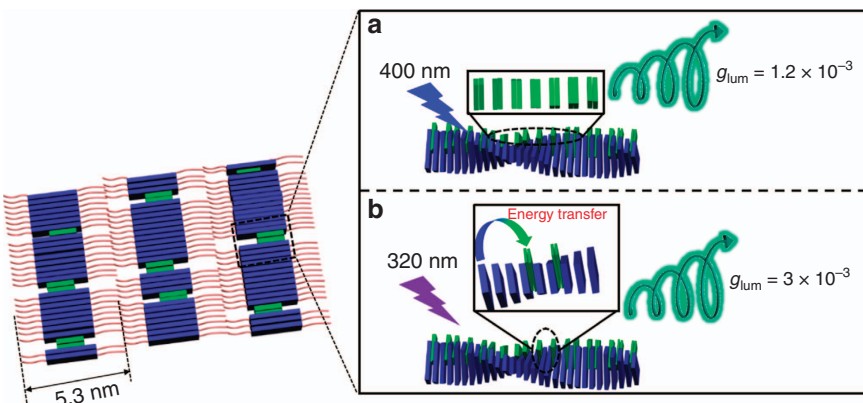

**Figure 6 | Schematic diagram of energy transfer amplified CPL in composite nanohelix.** (**a**) The composite nanohelix showed green circularly polarized emission with $g_{lum} = 1.2 \times 10^{-3}$ excited by 400 nm light; (**b**) energy transfer amplified CPL with a relative large value $g_{lum} = 3 \times 10^{-3}$ excited by 320 nm ultraviolet light.

excitation spectrum of L-1/BPEA monitored at 530 nm (Supplementary Fig. 18a). The excitation spectrum monitored at 530 nm resembled to the absorption of both donor and acceptor, which convince us that the energy transfer occurred in this system. In supramolecular hybrid system, according to the phase separation of acceptor BPEA, the excited energy from L-1 cannot exclusively transfer to BPEA. Therefore, the excitation spectrum of L-1/BPEA co-gel by monitoring at 530 nm would always exhibit the characteristic peak of acceptor BPEA.

To reveal the role of energy transfer in the induced CPL, we have further compared the CPL spectra of L-1(D-1)/BPEA co-gel by exciting the donor L-1(D-1) at 320 nm and directly exciting the acceptor BPEA at 400 nm. We had clearly observed the energy transfer amplified CPL by comparing the CPL spectra excited at 320 or 400 nm. As shown in Fig. 5c, the observed CPL dissymmetry factor $g_{lum}$ by exciting at 320 nm exhibited larger values than the one by directly exciting the BPEA at 400 nm. This clearly indicated that energy transfer from L-1(D-1) to BPEA could amplify the CPL of BPEA, which we name it as ETACPL. Typically, the dissymmetric factor of the CPL spectra at peak 500 nm by exciting at 400 nm are $\pm 1.2 \times 10^{-3}$, which are less than half of that by exciting at 320 nm ($g_{lum} = \pm 3 \times 10^{-3}$). Thus, energy transfer could significantly amplify the dissymmetric effect. The chiral energy transfer occurred in the self-assembled nanohelix, so it is interesting to investigate the effect of length of helix structures on amplification efficiency. To address this question, we studied the effect of sonication treatment on the morphology and CPL spectra of the D-1/BPEA gel. As shown in Supplementary Fig. 19, the helical nanofibre for the original sample was long and entangled with each other. After sonication treatment for 20 min, the gel collapsed and became suspension. From the SEM image, the length of fibres after sonication decreased remarkably but the helical pitch of the nanohelix did not show observable variation. It is well known that the supramolecular gelation is supported by the ordered nanostructures formed by low-molecular weight organic molecules. The shortened length of fibre at the microscale is consistent with the sonication-induced gel collapse at the macroscale. It indicated that the sonication treatment caused some damage to the self-assembly structure. Further prolonging the sonication treatment to 40 and 60 min, the length of fibres did not change obviously. We further measured the CPL spectra of D-1/BPEA gel after different sonication treatment. As shown in Supplementary Fig. 20, after different ultrasonic time, all the samples exhibited negative peak at 500 nm. The calculated CPL dissymmetry factors $g_{lum}$ at 500 nm for the sample sonicated with 20 min are

$-5.9 \times 10^{-4}$ ($\lambda_{ex} = 320$ nm), $-3.5 \times 10^{-4}$ ($\lambda_{ex} = 400$ nm). The dissymmetry factors $g_{lum}$ at 500 nm for 40 and 60 min ultrasonic time are $-8.6 \times 10^{-4}$ ($\lambda_{ex} = 320$ nm), $-6 \times 10^{-4}$ ($\lambda_{ex} = 400$ nm) and $-7.4 \times 10^{-4}$ ($\lambda_{ex} = 320$ nm), $-3 \times 10^{-4}$ ($\lambda_{ex} = 400$ nm), respectively. From the CPL data, two points should be stressed. First, the dissymmetry factors $g_{lum}$ for the samples undergoing sonication treatment showed $\sim 1$ order of magnitude less than the original sample (from $10^{-3}$ to $10^{-4}$). This suggested that sonication reduced the length of helix structures and induced some disruption of the well-ordered helical arrangement of the asemblies. Second, the dissymmetry factor $g_{lum}$ excited at 320 nm is still larger than the one excited at 400 nm. Considering that the energy transfer contributed to the dissymmetry factor $g_{lum}$ excited at 320 nm but not to the $g_{lum}$ excited at 400 nm, the D-1/BPEA gel after sonication still exhibited ETACPL. This supporting the general phenomenon on the energy transfer amplified CPL. So far, many of the chiral and CPL system have been reported, in which the chirality as well as the CPL are directly from the chiral molecules or supramolecular chiral transfer. Here we have found that the achiral acceptor could both capture the supramolecular chirality and collect the circularly polarized energy from the chiral assemblies, showing both supramolecular chirality and ETACPL. This affords us an excellent example of the multichannel chiroptical information communication in supramolecular system.

**Discussion**

From the data above, we have observed both chirality and energy transfer in the chiral nanosystem, an underlined mechanism for such transfer illustrated in the Fig. 6. The chiral gelator self-assembled into an ordered lamellar structure, in which the molecular monolayer served as the basic unit, which further formed a multilayer and rolled into nanohelix due to the driving force of molecular chiral centre. The chirality can be transferred from a molecular level to the nanoscale level, where the hydrogen bond between the amide groups, as well as the π–π stacking played important roles. The molecular packing and the non-covalent interactions could be confirmed by the XRD and FTIR measurements, respectively. Due to the AIE effect, the nanohelix showed remarkable assembly enhanced luminescence as well as CPL[53].

On the other hand, when mixed with achiral acceptors, the π–π interaction between the donor and acceptor caused the insertion of the acceptor molecules into the nanohelix. Such interaction maintained the nanohelix but caused the further chirality transfer, as illustrated in Fig. 6. In this case, although the handedness of the

nanohelix did not change, the packing of the acceptor is in the opposite direction of the nanohelix, therefore, we observed inversion of the supramolecular chirality localized on the BPEA. It is interesting that the guest (acceptor) molecules can carry on either the same or the opposite chiral packing against the host (donor) chiral matrices, which have been widely reported in supramolecular-assembled chiral systems[5,56–58]. This phenomenon has been explained to the different molecular packing between host chiral donors and guest achiral acceptors according to the theory of exciton-coupled circular dichroism[59].

Due to the energy transfer, the D–A composite assemblies showed luminescence from BPEA unit. In addition, since the chromophores aligned in a helical way, it emitted circularly polarized light. Furthermore, since the alignment of BPEA is in an opposite direction to the host chiral gelator, it also showed inversed CPL. Remarkably, the CPL through the energy transfer showed amplified intensity. In particularly, more than 2.5 times amplified luminescence dissymmetry factor $g_{lum}$ is observed for the composite nanohelix. Considering that $g_{lum}$ is a non-dimensional parameter, these results indicate that the energy transfer could amplify the intrinsical luminescence dissymmetry. This might be due to the BPEA emission enhancement via the energy transfer, which seems to further lead to the amplification of the $g_{lum}$ values. So far, there are several reported cases about CPL emitted from the achiral molecules in the chiral matrix[26,51,60]. Here we showed an example that induced CPL was amplified through energy transfer.

In summary, we have realized ETACPL system that is established by chiral donor and achiral acceptor. It was revealed that the host gelator self-assembled into chiral nanohelix showing strong supramolecular chirality as well as CPL. Such nanohelix could further transfer the chirality to an achiral acceptor through weak π–π interaction and co-assembly into the nanohelix. Remarkably, the chiral gelator assemblies could transfer the excited energy, which is a circularly polarized light, to the acceptor, leading to the energy transfer amplified CPL. The work demonstrated that the multichannel communications, such as chirality and energy, will provide deep insight into the designing functional chiroptical materials.

## Methods

**Materials.** All reagents and solvents were used as received otherwise indicated. Milli-Q water (18.2 MΩcm) was used in all cases. 4-(cyanomethyl) benzoate and methyl terephthaladehydate were purchased from TCI. BPEA was purchased from TCI and used as received. The energy donor gelators were synthesized by following the method reported previously. Cyano-substituted chromophore was synthesized according to the reported procedures. By introducing carboxylic acid moieties to the cyano-substituted chromophore, amide condensation reaction with glutamate-based amine was carried on using 1-ethyl-3-(3-dimethyllaminopropyl)carbodiimide hydrochloride (EDC•HCl)/1-hydroxybenzotrizole (HOBt) condensation agent. All the gelators were purified by column chromatography and confirmed the molecular structures by $^1$H NMR, MALDI-TOF-MS and elemental analysis.

**Characterization.** The $^1$H NMR spectra were recorded on a Bruker AV400 (400 MHz) spectrometer. Mass spectral data were obtained by using a BIFLEIII matrix-assisted laser desorption/ionization time of fight mass spectrometry (MALDI-TOF-MS) instrument. Elemental analysis was performed on a Carlo–Erba 1106 Thermo-Quest. UV–vis, CD and LD spectra were obtained using Hitachi UV-3900 and JASCO J-810 spectrometers, respectively. CPL measurements were performed with a JASCO CPL-200 spectrometer. Cuvettes of 1 mm were used for measuring the UV–vis, and FL spectra of samples. Cuvettes of 0.1 mm were used for measuring the CD and CPL spectra. For the measurement of CD spectra, the cuvette was placed perpendicularly to the light path of the CD spectrometer and rotated within the cuvette plane, to rule out the possibility of birefringence phenomena and eliminate the possible angle dependence of the CD signals. To estimate the contribution of LD effect on the true CD signal, 36 CD and LD spectra of the gel were measured in steps of 10° by rotating the sample in 1 mm cuvette fixed in the homemade rotator. Fluorescence spectra were recorded on a Hitachi F-4600 fluorescence spectrophotometer. XRD analysis was performed on a Rigaku D/Max-2500 X-ray diffractometer (Japan) with CuKa radiation (λ = 1.5406 Å), which was operated at a voltage of 40 kV and a current of 200 mA. FTIR studies were performed

with a JASCO FTIR-660 spectrometer. SEM was performed on a Hitachi S-4800 FE-SEM with an accelerating voltage of 10 kV. Before SEM measurements, the samples on silicon wafers were coated with a thin layer of Pt to increase the contrast. Fluorescent microscopy was recorded on the Olympus FV1000-IX81 confocal microscope system with × 100 oil immersion objective, using high-pressure mercury lamp as excitation source for fluorescent images. The absolute fluorescence quantum yield was measured by using an absolute PL quantum yield spectrometer (Edinburg FLS-980 fluorescence spectrometer) with a calibrated integrating sphere and fluorescence lifetime measurements were recorded on the same spectrometer using time-correlated single photon counting. The rheological properties of the gel were measured at 25 ± 0.05 °C with a Thermo Haake RS300 rheometer (cone and plate geometry of 40 mm in diameter).

**Data availability.** The data that support the findings of this study are available from the corresponding author on request.

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

## Acknowledgements

This work was supported by Basic Research Development Program (2013CB834504 and 2016YFA0203400) of the National Natural Science Foundation of China (Nos. 91027042 and 51673050), Strategic Priority Research Program of the Chinese Academy of Sciences (XDB12020200) and New Hundred-Talent Program research fund of the Chinese Academy of Sciences.

## Author contributions

D.Y. synthesized the samples and carried out all the characterizations, L.Z., M.L. and P.D. supervised the work. All authors participated in the discussion and interpretation of the results and co-wrote the manuscript.

## Additional information

**Competing interests:** The authors declare no competing financial interests.

 