## [Peer Review File · Nature Communications]

Reviewers' comments:

Reviewer #1 (Remarks to the Author):

This is an excellent study on coupling of chiral transfer and energy transfer. The authors made significant efforts to collect high-level experimental data. I basically recommend publication of this work in Nature Communications. I have the following major and minor requests (questions)

1) Major question

Although discussions on the data interpretation is really interesting and meaningful, details of chiral-energy transfer is not completely convincing.

(A) Possible mechanisms of chiral energy transfer

(A-1) Single chiral donor molecule to single acceptor molecule via energy transfer (single to single)

(A-2) Fixing single acceptor molecule in chiral environments of helix creates chiral emission via energy transfer (multi to single)

(A-3) Fixing acceptor molecules on the helical host structure induce chiral array resulting in chiral emission via energy transfer (multi to multi).

Amplification of chiral energy transfer may come from (A-3), but it is not fully clear.

(B) What is requirement of supramolecular chirality for chiral energy transfer? What is difference from simple energy transfer? What is necessary length oh helical structures?

In order to answer to these questions, it would be a good idea to investigate effect of length of helix structures on amplification efficiency. For example, experiments with broken or fragmented assemblies (broken by sonication etc) would give results interesting for discussion on requirements of supramolecular chirality effects and mechanisms.

2) Minor revision.

Reference selection is very good from various research groups, but recent comprehensive reviews on chiral interaction and supramolecular assemblies had better be added more (for example, Acc. Chem. Res. 48, 521-529 (2015), Bull. Chem. Soc. Jpn. 89, 1277-1306 (2016), Chem. Rev. 116, 13752-13990 (2016)), because this area become very hot and new facts are appearing rapidly.

Reviewer #2 (Remarks to the Author):

The manuscript entitled "Chirality and Energy Transfer Amplified Circularly Polarized Luminescence in Composite Nanohelix Composed of Chiral Donor and Achiral Acceptor" deals with amplified circularly polarised luminescence obtained due to both the transfer of supramolecular chirality as well as energy between a cyano substituted stilbene derivative conjugated with a gelator moiety N,N'-bis(dodecyl)-L(D)-amine glutamic diamide (LGAm/DGAm) as donor and an achiral acceptor moiety 9,10-bis(phenylethynyl) anthracene (BPEA). Authors have also demonstrated efficient energy transfer in this D-A system along with the significant amplification of circularly polarised fluorescence in a self-assembled nanoscale system. The selection of a suitable solvent mixture was essential to strengthen the interaction of the heterogeneous assembly without obvious phase separation. Photophysical studies on the donor moiety and the assembled D-A system (effectively maintaining a particular ratio) exhibited distinct emission changes as a result of highly efficient (86%) energy transfer. The X-ray Diffraction pattern also supports the intactness of the system even after the assembly formation between the D and A. Finally, a detailed investigation on the enhanced CPL (almost 2.5 times) of the composite nanohelix formed due to n-n interaction between the donor and the inserted acceptor was discussed.

Specific Comments:

1) There is no rheological data regarding the gelation process. Authors should comment on the

strength of the gel before and after the insertion of the acceptor moiety.

2) Though authors have provided some evidence to prove energy transfer process, it is important to conduct detailed fluorescence lifetime studies to rule out the possibilities of trival (emission-reabsorption) mechanism. They may refer *Angew. Chem., Int. Ed.* 46, 6260–6265 (2007); *Adv. Mater.* 21, 2059–2063 (2009); *Phys. Chem. Chem. Phys.* 13, 4942–4949 (2011).

3) Authors should recast the following sentence: (a) Page 9: "Considering that serious stacking . . . n-conjugated luminescent systems" ; energy quenching?; (b) Page 11: X-ray diffraction (XRD) spectroscopy ?.

4) Page 11: The authors should discuss the XRD data in detail by correlating the results with molecular parameters and packing.

5) I feel the following reviews related to light harvesting gels and supramolecular chirality relevant in the context of the present paper can be cited. *Chem. Soc. Rev.* 37, 109–122 (2008); *Angew. Chem., Int. Ed.* 53, 365–368 (2014); *Chem. Soc. Rev.* 43, 4222–4242 (2014); *Angew. Chem., Int. Ed.* 46, 8948–8968 (2007); *Bull. Chem. Soc. Jpn.* 81, 1196–1211 (2008).

This work will be of interest to broad community of scientists. The manuscript can be considered for publication once authors address the above-mentioned points.

Reviewer #3 (Remarks to the Author):

Liu et al reported in the paper a tandem photoexcited resonance energy (possibly FRET) and chirality transfer as proven by circularly polarized luminescent signal (CPL) in binary molecular-based nano-helical aggregate system. The system consists of chiral FRET donor and achiral FRET acceptor. The CPL arises from the achiral FRET acceptor, not from the chiral FRET donor. The work is well-organized by convincing with supporting information and publishable. But I kindly ask the authors to reply my naive queries and comments on behalf of suspicious readers and specialists over the world. This is because *NComm* is one of the influential journals. Indeed, I find many critical public comments from the world-wide readers and specialists if any papers at Nature's sister journals opened once. Recently, I knew several public comments to some paper(s) appeared in *Nature Materials* (NM) and *Scientific Reports* (SP). Eventually, the authors of NM had to retract his/her paper by the public comment(s), due to a scientific misconduct or a fatal misleading results/analysis or due to instrumental errors.

1. First of all, do you think of what kinds of energy transfer, Förster (so-called FRET) or Dexter (tunneling type) or other type? Should mention this.

2. If this was FRET type system, should evaluate a FRET distance between the donor and the acceptor. You already showed Stern-Volmer type plot (FigS4). This plot may help you.

3. Should illustrate Jablonski energy diagram including S₀, S₁, and S₂ states of the chiral donor and achiral acceptor along with Kasha's rule. Recently, anti-Kasha's rule is becoming a research topic.

4. Should explain the origin of bisignate (split type noted in manuscript) CD band in the range of 250-600 nm (Fig4a). Should show both UV-vis spectra of L-1 and D-1 associated with the corresponding CD spectra.

5. Explain the origin of bisignate induced CD band (ICD) in the range of 440-500 nm (Fig4b).

6. Which chirality at the L-1/D-1 or helicity of the L-1/D-1 nano helix is responsible for the ICD of BPEA? Helicity based on stereogenic bonds and local chirality based on stereogenic centers often contribute to oppositely induce the resulting chiroptical sign.

7. "their CD spectrum" should be their CD spectra (pls check again other sl and pl as countable nouns)

8. Two positive signals at 445 and 470 nm. I think, this is as a consequence of a bisignate CD band due to chirally assorted origin, so-called exciton couplet. I estimated the gabs values at 445/470nm are ± 0.02 , respectively, for L-1 with BPEA. CPL at 500 nm along with weak vibronic CPLs at 520 nm is from the mixed gel 470 nm band. The CD sign at 470 nm is identical to that of CPL at 410 nm. Based on Stokes shift between the ground-state and photoinduced reorganization of emitters at the S1-state, if Kasha's rule is applied. Actually, for L-1 alone gel CD sign at 380 nm is identical to that of CPL sign at 500 nm.

9. It is unclear for me, where BPEA is placed to L-1/D-1 nano helix, whether it is inside or outside and whether isolated or assorted already. According to illustration of Scheme 1, BPEA is located into nano helix as isolated molecules, But the corresponding CD band of BPEA originates from the exciton couplet as I mentioned above, possibly due to chirally assorted BPEA molecules, possibly, not due to isolated BPEA.

10. g_{lum} at 410 nm $\pm 1.1 \times 10^{-2}$, g_{lum} at 510-520 nm $\mp 0.3 \times 10^{-2}$ (Fig5b, inset) but g_{lum} at 510-520 nm $\mp 3 \times 10^{-2}$ (Fig5c, plot). Pls confirm this inconsistency.

11. Pls disclose the corresponding PL signal simultaneously obtained the CPL spectra in Fig. 5.

12. In Fig S3b, PL spectrum of the BPEA film is strange for me. The normalized PL spectra revealed three major bands at 260, 610, and 640 nm. I assume the 640 nm band with a very narrow bandwidth of ≈ 10 nm is due to stray light when excited at 320 nm with a very narrow bandwidth of ≈ 5 nm due to ill-tuned PL instrumental origin. For example, the damaged grating or broad bandwidth for excitation, multiple scattering at the interface of air/film/substate often cause this stray light. Should re-measure PL using a specific cut-filter or moving excitation wavelength from 320 nm to several other wavelengths (300, 310, 330, 340nm) whether the 640 nm narrow PL band remains or retains or moves to blue or red associated with these exception wavelengths.

13. Should disclose in Fig S3a, show excitation, dichroic, and lang-pass filtered wavelengths for obtaining the image.

14. In my computer, some messy codes and no fonts, possibly, symbol, nu and delta, are displayed. Should re-check manuscript.

Response to the comments of reviewers

To Reviewer #1:

This is an excellent study on coupling of chirality transfer and energy transfer. The authors made significant efforts to collect high-level experimental data. I basically recommend publication of this work in Nature Communications. I have the following major and minor requests (questions)

1) Major question

Although discussions on the data interpretation is really interesting and meaningful, details of chiral-energy transfer is not completely convincing.

(A) Possible mechanisms of chiral energy transfer

(A-1) Single chiral donor molecule to single acceptor molecule via energy transfer (single to single)

(A-2) Fixing single acceptor molecule in chiral environments of helix creates chiral emission via energy transfer (multi to single)

(A-3) Fixing acceptor molecules on the helical host structure induce chiral array resulting in chiral emission via energy transfer (multi to multi).

Amplification of chiral energy transfer may come from (A-3), but it is not fully clear.

(B) What is requirement of supramolecular chirality for chiral energy transfer? What is difference from simple energy transfer? What is necessary length of helical structures?

In order to answer to these questions, it would be a good idea to investigate effect of length of helix structures on amplification efficiency. For example, experiments with broken or fragmented assemblies (broken by sonication etc) would give results interesting for discussion on requirements of supramolecular chirality effects and mechanisms.

Author reply: Thank you very much for your valuable suggestions. As you said, chirality transfer mechanism from chiral host molecule to achiral dopant could be summarized as three typical modes: single to single, multi to single and multi to multi. As for the case of single to single, it needs strong interaction such as electrostatic interaction or complex formation between the chiral host and achiral dopant molecules. Chirality transfer from multi to single and multi to multi can always exist in the self-assembled system. In these two cases, the weak noncovalent interactions such as hydrogen bond and π - π stacking play important roles as the driving force for chirality transfer. In our present study, there was no clear evidence for the complex formation. The possible channel for chirality transfer is weak π - π interaction between the chiral gelator (L-1 or D-1) and achiral BPEA molecule. Thus, the chirality transfer process can only occur through multi to single or multi to multi. Furthermore, in the CD spectra of L-1/BPEA and D-1/BPEA, as shown in Figure 4b, the corresponding bisignate CD band of BPEA in the range of 440-500 nm originating from the exciton couplet indicated that BPEA molecules existed as aggregates embedded in the nanohelix. Thus, the chirality transfer mechanism should be multi to

multi. On the other hand, the energy transfer from donor to acceptor was also through multi to multi mechanism. The maximum energy transfer efficiency was estimated with a relative high value of 86% in the assemblies. In fact, the energy transfer from donor (L-1 or D-1) to BPEA acceptor can also occur in the non-assembly state (single to single), as shown in Figure S7. But the maximum energy transfer efficiency was only about 65% in dilute CHCl_3 solution. Thus, the chiral energy transfer can be explained by the multi to multi mechanism.

Figure. R. 1. SEM images of xerogel made from D-1/BPEA=5/1 with different ultrasonic time: a) 0 min, b) 20 min, c) 40 min, d) 60 min.

Figure. R. 2. CPL spectra of D-1/BPEA=5/1 gel with different ultrasonic time: a) 20 min, b) 40 min, c) 60 min. The samples were excited with 320 nm (black line) and 400 nm (red line) respectively. The ultrasonic treatment was conducted in KH5200E instrument and the power was kept constant at 200 W. The calculated CPL dissymmetry factors g_{lum} at different ultrasonic time are as follow: a) -5.9×10^{-4} ($\lambda_{ex} = 320$ nm), -3.5×10^{-4} ($\lambda_{ex} = 400$ nm); b) -8.6×10^{-4} ($\lambda_{ex} = 320$ nm), -6×10^{-4} ($\lambda_{ex} = 400$ nm); c) -7.4×10^{-4} ($\lambda_{ex} = 320$ nm), -3×10^{-4} ($\lambda_{ex} = 400$ nm).

In order to investigate the effect of length of helix structures on amplification efficiency, we studied the effect of sonication treatment on the morphology and CPL spectra of the D-1/BPEA gel. As shown in Figure R1, the helical nanofiber for the original sample was long and entangled with each other. After sonication treatment for 20 minutes, the gel collapsed and became suspension. From the SEM image, the length of fibers after sonication decreased remarkably but the helical pitch of the nanohelix didn't show observable variation. It is well known that the supramolecular gelation is supported by the ordered nanostructures formed by low-molecular weight organic molecules. The shortened length of fiber at the microscale is consistent with the sonication-induced gel collapse at the macroscale. It indicated that the sonication treatment caused some damage to the self-assembly structure. Further prolonging the sonication treatment to 40 and 60 minutes, the length of fibers didn't changed obviously. We further measured the CPL spectra of D-1/BPEA gel after different sonication treatment. As shown in Figure. R2, after different ultrasonic time, all the samples exhibited negative peak at 500 nm. The calculated CPL dissymmetry factors g_{lum} at 500 nm for the sample sonicated with 20 minutes are -5.9×10^{-4} ($\lambda_{ex} = 320$ nm), -3.5×10^{-4} ($\lambda_{ex} = 400$ nm). The dissymmetry factors g_{lum} at 500 nm for 40 and 60 minutes ultrasonic time are -8.6×10^{-4} ($\lambda_{ex} = 320$ nm), -6×10^{-4} ($\lambda_{ex} = 400$ nm) and -7.4×10^{-4} ($\lambda_{ex} = 320$ nm), -3×10^{-4} ($\lambda_{ex} = 400$ nm) respectively. From the CPL data, two points should be stressed. Firstly, the dissymmetry factors g_{lum} for the samples undergoing sonication treatment showed ~ 1 order of magnitude less than the original sample (from 10^{-3} to 10^{-4}). This suggested that sonication reduced the length of helix structures and induced some disruption of the well-ordered helical arrangement of the assemblies. Secondly, the dissymmetry factor g_{lum} excited at 320 nm is still larger than the one excited at 400 nm. Considering that the energy transfer contributed to the dissymmetry factor g_{lum} excited at 320 nm but not to the g_{lum} excited at 400 nm, the D-1/BPEA gel after sonication still exhibited energy transfer amplified circularly polarized luminescence. This supporting the general phenomenon on the energy transfer amplified CPL.

Here, the requirement of supramolecular chirality for chiral energy transfers needs the formation of the collective chiral system and there is an overlap between donor emission and absorption of the acceptor. For simple energy transfer, it is no need that donor and acceptor in a chiral system.

2) Minor revision.

Reference selection is very good from various research groups, but recent comprehensive reviews on chiral interaction and supramolecular assemblies had better be added more (for example, Acc. Chem. Res. 48, 521-529 (2015), Bull. Chem. Soc. Jpn. 89, 1277-1306 (2016), Chem. Rev. 116, 13752-13990 (2016)), because this area become very hot and new facts are appearing rapidly.

Author reply: Thanks very much for your kind suggestions. We have read the comprehensive reviews carefully as you mentioned. In our revised manuscript, we have added these important references.

Answer to Reviewer #2

The manuscript entitled “Chirality and Energy Transfer Amplified Circularly Polarized Luminescence in Composite Nanohelix Composed of Chiral Donor and Achiral Acceptor” deals with amplified circularly polarised luminescence obtained due to both the transfer of supramolecular chirality as well as energy between a cyano substituted stilbene derivative conjugated with a gelator moiety N,N'-bis(dodecyl)-L(D)-amine glutamic diamide (LGAm/DGAm) as donor and an achiral acceptor moiety 9,10-bis(phenylethynyl) anthracene (BPEA). Authors have also demonstrated efficient energy transfer in this D-A system along with the significant amplification of circularly polarised fluorescence in a self-assembled nanoscale system. The selection of a suitable solvent mixture was essential to strengthen the interaction of the heterogeneous assembly without obvious phase separation. Photophysical studies on the donor moiety and the assembled D-A system (effectively maintaining a particular ratio)

exhibited distinct emission changes as a result of highly efficient (86%) energy transfer. The X-ray Diffraction pattern also supports the intactness of the system even after the assembly formation between the D and A. Finally, a detailed investigation on the enhanced CPL (almost 2.5 times) of the composite nanohelix formed due to π - π interaction between the donor and the inserted acceptor was discussed.

Specific Comments:

1) There is no rheological data regarding the gelation process. Authors should comment on the strength of the gel before and after the insertion of the acceptor moiety.

Author reply: Thanks very much for your kind suggestion. As you suggested, we have performed stress-sweep rheological measurement to study the strength of the gel before and after the insertion of the acceptor BPEA. As shown in Figure. R3, the storage modulus (G') of the single-component D-1 gel proved to be ~ 1.5 orders of magnitude larger than the loss modulus (G''), which is consistent with solid-like behavior. While in the D-1/BPEA two-component gel, the storage modulus (G') of the single-component D-1 gel proved to be ~ 2.5 orders of magnitude larger than the loss modulus (G''). Obviously, the acceptor BPEA reinforced the magnitude of the storage modulus. Further, the presence of BPEA also resulted in increase of magnitude in the yield stress. These observations indicate that the insertion of the acceptor moiety lead to enhancement in the strength of the gel.

Figure. R.3. Stress sweep rheology of the single-component D-1 gel ($[D-1] = 2 \text{ mM}$) and two-component D-1/BPEA gel ($[D-1] = 2 \text{ mM}$, $[BPEA] = 0.1 \text{ mM}$). The measurement was conducted at constant $25 \text{ }^\circ\text{C}$ and the measurement frequency was fixed as 0.1 Hz .

2) Though authors have provided some evidence to prove energy transfer process, it is important to conduct detailed fluorescence lifetime studies to rule out the possibilities of trival (emission-reabsorption) mechanism They may refer *Angew. Chem., Int. Ed.* 46, 6260–6265 (2007); *Adv. Mater.* 21, 2059–2063 (2009); *Phys. Chem. Chem. Phys.* 13, 4942–4949 (2011).

Figure. R. 4. Emission decay curves of the D-1 and D-1 / BPEA = 5/1 gels monitored at 450 nm . $\lambda_{\text{exc}} = 358.4 \text{ nm}$, IRF= instrument response function, $[D-1] = 2\text{mM}$, $[BPEA] = 0.4 \text{ mM}$.

Author reply: Thanks very much for your kind comment. The energy transfer mechanisms usually include trivial mechanism, Förster mechanism and Dexter mechanism. We conducted the fluorescence lifetime measurement to study the possible mechanism (*Angew. Chem., Int. Ed.* 46, 6260–6265, 2007). As shown in Figure. R. 4, in the absence of BPEA acceptor, the D-1 gel exhibited triple exponent decay with time constants of $\tau_1 = 0.48 \text{ ns}$ (71%), $\tau_2 = 2.06 \text{ ns}$ (18%), $\tau_3 = 10.01 \text{ ns}$ (11%). The calculated average time of τ was about 1.15 ns . However, the emission decay of D-1 becomes faster in the presence of BPEA. When 20 mol% BPEA was

added, D-1 exhibited a fast triple exponent decay with time constants of $\tau_1 = 0.43$ ns (71%), $\tau_2 = 1.39$ ns (22%), $\tau_3 = 7.02$ ns (7%) and the calculated average time of τ was about 0.52 ns. The shortening of the averaged emission decay time of the D-1 donor in the presence of acceptor BPEA indicated that Förster mechanism may behave as the major mechanism for energy transfer. We have added this part into our revised manuscript. The mentioned references were also cited in appropriate position.

3) Authors should recast the following sentence: (a) Page 9: “Considering that serious stacking . . . π -conjugated luminescent systems” ; energy quenching?; (b) Page 11: X-ray diffraction (XRD) spectroscopy ?.

Author reply: Thanks very much for your detailed reading and kind suggestion. After deliberate consideration, we have changed the sentence “Considering that serious stacking induced energy quenching often occurred in π -conjugated luminescent system, L-1 is an outstanding energy donor” in page 9, and the more appropriate expression is “Considering that π - π stacking in aggregates always induced bathochromic shift of emission peak and quenched fluorescence emission related to the formation of excimer or exciplex in general π -conjugated luminescent systems, L-1 with stable emission peak position is a proper energy donor in the self-assembly system”. In page 11, “X-ray diffraction (XRD) spectroscopy” was changed to “X-ray diffraction measurement”. The changed sentences have replaced the original in our revised manuscript.

4) Page 11: The authors should discuss the XRD data in detail by correlating the results with molecular parameters and packing.

Author reply: Thank you very much for your kind suggestions. The calculated molecular length for L-1 (or D-1) is about 5.6 nm according to the CPK model. In the XRD spectrum for L-1 xerogel, two θ values appeared at 1.62, 3.2, 4.8° and so on. The corresponding distances were estimated to be 5.4, 2.7, 1.84 nm and so on according to the Bragg's equation. The d -spacing ratio is about 1:1/2:1/3, which is consistent with the lamellar structure. The calculated molecular length (5.6 nm) is approximately equal to the experimental date (5.4 nm). So the possible molecular packing pattern is shown as the Scheme 1 in the manuscript: the chiral gelator self-assembled into an ordered lamellar structure, in which the molecular monolayer served as the basic unit, which further formed a multilayers and rolled into nanohelix. For the L-1/BPEA xerogel, the XRD pattern was identical to the one of pure L-1. This confirmed that the L-1 molecules adopt the same packing pattern as the pure L-1 and the addition of BPEA to L-1 did not destroy the well-ordered packing of L-1. In the revised manuscript, we have included above discussions.

5) I feel the following reviews related to light harvesting gels and supramolecular chirality relevant in the contest of the present paper can be cited. Chem. Soc. Rev. 37, 109–122 (2008); Angew. Chem., Int. Ed. 53, 365–368 (2014); Chem. Soc. Rev. 43, 4222–4242 (2014); Angew. Chem., Int. Ed. 46, 8948–8968 (2007); Bull. Chem. Soc. Jpn. 81, 1196–1211 (2008).

Author reply: Thanks very much for your kind suggestions. We have carefully read the reviews related to light harvesting and supramolecular chirality as you advised. We found these studies are very innovative and instructive for our future research. We have cited these reviews in our revised manuscript.

Reviewer #3 (Remarks to the Author):

Liu et al reported in the paper a tandem photoexcited resonance energy (possibly FRET) and chirality transfer as proven by circularly polarized luminescent signal (CPL) in binary molecular-based nano-helical aggregate system. The system consists of chiral FRET donor and achiral FRET acceptor. The CPL arises from the achiral FRET acceptor, not from the chiral FRET donor. The work is well-organized by convincing with supporting information and publishable. But I kindly ask the authors to reply my naive queries and comments on behalf of suspicious readers and specialists over the world. This is because NComm is one of the influential journals. Indeed, I find many critical public comments from the world-wide readers and specialists if any papers at Nature's sister journals opened once. Recently, I knew several public comments to some paper(s) appeared in Nature Materials (NM) and Scientific Reports (SP). Eventually, the authors of NM had to retract his/her paper by the public comment(s), due to a scientific misconduct or a fatal misleading results/analysis or due to instrumental errors.

1. First of all, do you think of what kinds of energy transfer, Förster (so-called FRET) or Dexter (tunneling type) or other type? Should mention this.

Author reply: Thanks very much for your comment. The energy transfer mechanisms usually include trivial mechanism, Förster mechanism and Dexter mechanism. When the excited donor meet the ground state of acceptor, energy transfer may occur with only one mechanism, or it might occur with multiple mechanisms. We conducted the fluorescence lifetime measurement to study the possible mechanism. As shown in Figure. R.4, in the absence of BPEA acceptor, the D-1 gel exhibited triple exponent decay with time constants of $\tau_1 = 0.48$ ns (71%), $\tau_2 = 2.06$ ns (18%), $\tau_3 = 10.01$ ns (11%). The calculated average time of τ was about 1.15 ns. However, the emission decay of D-1 becomes faster in the presence of BPEA. When 20 mol% BPEA was added, D-1 exhibited a fast triple exponent decay with time constants of $\tau_1 = 0.43$ ns (71%), $\tau_2 = 1.39$ ns (22%), $\tau_3 = 7.02$ ns (7%) and the calculated average time of τ was about 0.52 ns. The shorting of the averaged emission decay time of the D-1 donor in the presence of BPEA acceptor indicated that Förster mechanism may behave as the major mechanism for energy transfer (*Angew. Chem., Int. Ed.* 46, 6260–6265, 2007).

2. If this was FRET type system, should evaluate a FRET distance between the donor and the acceptor. You already showed Stern-Volmer type plot (FigS4). This plot may help you.

Author reply: Thank you very much for your kind suggestions. In FRET type system, the FRET distance (abbreviated as R_0) is defined as that donor-acceptor separation where 50% of the donor energy is transferred to the acceptor. The value of R_0 can be calculated experimentally on the basis of the following equation:

$$R_0 = [(8.79 \times 10^{-25}) (\kappa^2 / N^4) \phi_D J_{AD}]^{1/6}$$

In this equation, κ^2 denotes relative orientation between the transition dipoles of the donor and the acceptor, ϕ_D is the quantum yield of the donor molecule in the absence of the acceptor, N represents the refractive index of the medium and $J_{AD} = \int_0^\infty F_D(\lambda)\varepsilon(\lambda)\lambda^4 d\lambda$ where $F_D(\lambda)$ is the normalized fluorescence intensity of the donor and $\varepsilon(\lambda)$ denotes the molar extinction coefficient of the acceptor. In our system, the calculated overlap integral $J_{AD} = 3.271 \times 10^{14} \text{ nm}^4 \text{ M}^{-1} \text{ cm}^{-1}$, the refractive index of the medium is 1.46 and the quantum yield of L-1 gel formed by the donor molecule in the absence of the acceptor BPEA is 5%. However, the orientation relative orientation κ^2 between the transition dipoles of the donor and the acceptor is difficult to determine experimentally, and the uncertainties may lead to ambiguity in the determination of the FRET distance. Depending upon the relative orientation of donor and acceptor, κ^2 can be ranging from 0 to 4. For collinear and parallel transition dipoles, $\kappa^2 = 4$, and for parallel dipoles, $\kappa^2 = 1$. It has been reported that in highly viscous solution of peptides, where the orientations of donor and acceptor may be assumed to be random but fixed, the value chosen for $\kappa^2 = 0.476$ is appropriate. (*Annu. Rev. Biochem.*, 1971, 40, 83-114). These orientations do not change during the lifetime of the excited state. We think using the $\kappa^2 = 0.476$ in our system is reasonable to some degree. Furthermore, the sixth root of κ^2 is taken in calculating the distance, so variation of κ^2 results in only small changes in R_0 . Thus, we use $\kappa^2 = 0.476$ to evaluate the FRET distance. According to the above equation, the final calculated value of R_0 is 27.9 Å, which is in the range of 10 – 100 Å where FRET can occur effectively.

As for your suggestion, we also try to use the Stern-Volmer type plot to evaluate the FRET distance. We have calculated the FRET distance by using the following equation:

$$A_0 = 447 / R_0^3$$

Where A_0 is called the critical concentration, and represents the acceptor concentration that results in 76% energy transfer (Principles of fluorescence spectroscopy. Springer US, 1999). The concentration of A_0 is in moles per liter (M) and the R_0 is in units of angstroms. According to the Stern-Volmer type plot shown in Figure S4 in the original manuscript, the acceptor concentration that results in 76% energy transfer is about 2 mM / 10 = 2×10^{-4} M. Thus the calculated FRET distance $R_0 = 130$ Å, which is out of the range where FRET can occur effectively.

It should be noted that this equation is approximately accurate and suitable for the case that energy transfer occurs for donors and acceptors randomly distributed in three-dimensional solutions. Thus the acceptor concentrations need to be rather high for FRET between unlinked donor and acceptors in solution. However the high acceptor concentrations needed for FRET in solution will induce inner filter effects and make such measurements difficult. We think that it may produce serious error to use this equation to calculate the FRET distance in semitransparent supramolecular gel system with high acceptor concentration. Thus Stern-Volmer plotting is not suitable for the calculation of the FRET distance in our gel system.

3. Should illustrate Jablonski energy diagram including S_0 , S_1 , and S_2 states of the chiral donor and achiral acceptor along with Kasha's rule. Recently, anti-Kasha's rule is becoming a research topic.

Author reply:

Figure. R. 5. a) UV-vis spectrum for D-1 gel in DMSO/H₂O=9/1. [D-1] = 2 mM. 0.1 mm cuvettes were used for the measurement. This data comes from Figure 4 in the original manuscript. b) UV-vis spectrum for 3.1×10^{-4} M D-1 solution in cyclohexane. 10 mm cuvettes were used for the measurement. c) FL spectra for D-1 gel in DMSO/H₂O=10/1 at different excitation wavelength including 220 nm, 269 nm and 330nm. 1 mm cuvettes were used for the measurement. The emission slit, excitation slit and PMT voltage are 5 nm, 5 nm, 700 V. In Figure c, the 440 nm and 538 nm lights which are twice wavelength of the corresponding excitation lights come from the scattering effect.

Answer: Thanks very much for your thoughtful suggestion. Kasha's rule is a ubiquitously fundamental principle in photophysics and photochemistry. It describes the phenomenon that the photoluminescence of molecules should proceed only from their lowest excited singlet (S_1) state though they may be excited to any set of higher energy electronic states (denoted S_n , $n > 1$) from their electronic ground singlet state. The correctness of Kasha' rule is based on the fact that the rates of internal conversion, intersystem crossing, and vibrational relaxation from a given excited state to another excited state are much faster than the intrinsic radiative rate of that state. In steady state fluorescence, the emission band proceeds from the lowest excited energy states, being independent of the excitation wavelength. However, in some cases, the rates of emission decays are able to compete with internal conversion to the S_1 state. In this case, one will observe the appearance of characteristic short-wavelength bands in fluorescence (or phosphorescence) emission spectra at excitations to high-energy electronic states. This phenomenon is called anti-Kasha behavior.

In order to prove the validity of Kasha's rule in our system, we measured the absorption spectrum of the donor in gel state (DMSO/H₂O=9/1). As shown in Figure. R. 5a, the gel exhibited an absorption peak centered at 330 nm. According to the reports in literatures, the S_0 - S_1 absorption bands for cyano substituted stilbene are always located in the range of 310-360 nm. The absorption peak at 330 nm for D-1 gel should be also attributed to the S_0 - S_1 absorption. The further attempt to measure the possible higher transitions absorption (S_0 - S_2) in the gel was hampered by the

interference with the DMSO solvent in the range of 200-300 nm. So we have to choose another “transparent” solvent cyclohexane. The D-1 molecule can be dissolved in cyclohexane in very dilute concentration (3.1×10^{-4} M was used in the experiment for the proper absorption intensity). As shown in Figure. R. 5b, the D-1 molecule exhibited three absorption bands in the cyclohexane. The weak absorption in the range of 330-360 nm should be attributed to the S_0 - S_1 absorption. The other two absorption peaks at 220 nm and 269 nm in the short-wavelength area aroused our attention. We doubt whether one of them can be attributed to the higher transitions absorption (such as S_0 - S_2). In fact, it is hard to give a completely correct identification of the S_0 - S_2 transitions absorption just according to the steady absorption and emission spectra. The S_0 - S_2 transitions absorption in some cases is not well resolvable in the steady-state approach. In spite of the uncertainty for the attribution of two peaks at 220 nm and 269 nm, we still measured the emission spectra of the D-1 gel with different excitation wavelengths including 220 nm, 269 nm and 330 nm. As shown in Figure. R. 5c, wide emission band appeared at 420 nm region under different excitation. This emission band can only be from the S_1 - S_0 transitions. We can't observe any new emission peak in the short-wavelength region. This suggests that emission from chiral donor in the gel state is exclusively from the S_1 - S_0 transitions though they may be excited to any set of higher energy electronic states (220 nm and 269 nm may be). So we can conclude that our system obeys Kasha's rule. The Jablonski energy diagram including S_0 , S_1 states of the chiral donor and achiral acceptor along with Kasha's rule was shown in Figure. R.6.

Figure. R. 6. The Jablonski energy diagram for the chiral donor and achiral acceptor. When absorbing 320 nm light, the chiral donor was excited to higher vibrational level of either S_1 or S_2 , then rapidly relaxed to the lowest vibrational level of S_1 . The energy of donor at S_1 state was transferred to acceptor at S_0 state through resonant transitions. The acceptor at S_1 excited state then decayed to different vibrational level of S_0 state with 487 nm and 514 nm fluorescence emission. The S_0 - S_1 transition associated with 320 nm absorption for D-1 donor is expressed as 3.88 eV. The S_0 - S_2 transition associated with 312 nm absorption for BPEA acceptor is expressed as 3.97 eV. The

two emission bands located at 487 nm and 514 nm for BPEA are expressed as 2.55 eV and 2.41 eV.

4. Should explain the origin of bisignate (split type noted in manuscript) CD band in the range of 250-600 nm (Fig4a). Should show both UV-vis spectra of L-1 and D-1 associated with the corresponding CD spectra.

Author reply: Thank you very much for your suggestion. The bisignate CD band in the range of 240-440 nm comes from the exciton couplet of cyano-substituted stilbene chromophores in the supramolecular assemblies. According to the exciton-coupled circular dichroism theory, when two (or more) chromophores are located near in space and have a proper mutual orientation, their strong electric-dipole allowed transitions will couple to each other. The strong exciton coupling between the chromophores can result in the appearance of both a red-shifted band and a blue-shifted band with opposite signs in the absorption region of chromophores. The intensity and signals of the bisignate CD are decided by the distance and dihedral angle between the chromophores. A positive chirality arrange of the chromophores (when a clockwise rotation by an acute angle brings the dipole in the front onto that in the back) always results in the positive CD couplet and vice-versa.

In the L-1 (or D-1) gel system, the strong π - π stacking of the cyano-substituted stilbene chromophores is one of the driving forces for the self-assembly. On the other hand, the adjacent stilbene chromophores will arrange with some twisted angle due to the chiral bias of the attached chiral glutamate moieties. The strong exciton coupling between the stilbene chromophores will result to bisignate CD band in the 240 – 440 nm with a crossover with the X axis at 331 nm. The negative CD couplet for L-1 gel and positive CD couplet for D-1 gel indicated that the L-1 and D-1 molecules packed with left-handed and right-handed rotation mode, respectively.

We have also added the UV-vis spectra of L-1 and D-1 in their corresponding CD spectra. The UV-vis spectra of L-1 and D-1 gels have almost similar shape and absorption intensity.

5. Explain the origin of bisignate induced CD band (ICD) in the range of 440-500 nm (Fig4b).

Author reply: Thank you very much for your kind suggestions. The detailed discussion on the bisignate induced CD bands is shown in the part of reply 6. In fact, in the co-assembly of L-1/BPE (D-1/BPEA), π - π interaction between BPEA and stilbene chromophores attached to L-1(D-1) is the driving force for the co-assembling of BPEA and L-1(D-1). In the range of 440- 500 nm of the UV-vis spectra, the L-1/BPEA (D-1/BPEA) gel exhibited two absorption peaks at 445 nm and 470 nm, which can be attributed to the S_0 - S_1 transition with different vibronic peaks for BPEA aggregate molecules inserted into the assemblies. The corresponding two positive bisignate CD bands in the range of 440-500 nm (Figure 4b) for L-1/BPEA and two negative bisignate CD bands for D-1/BPEA indicated that the chirality transfer from the chiral donor to the achiral BPEA aggregate. It should be noted that the ICD signals

for BPEA are opposite to the bisignate CD signal of the corresponding chiral donor. This can be explained by the opposite helical arrangement of BPEA to the chiral donor in the co-assembly.

6. Which chirality at the L-1/D-1 or helicity of the L-1/D-1 nano helix is responsible for the ICD of BPEA? Helicity based on stereogenic bonds and local chirality based on stereogenic centers often contribute to oppositely induce the resulting chiroptical sign.

Author reply: Thank you very much for your thoughtful question. Supramolecular chirality generally arises from the spatial non-symmetric arrangement of molecules in a non-covalent assembly. In supramolecular system, the chiral assemblies or chiral molecules themselves can serve as the template to induce the achiral molecules to exhibit induced chirality. In fact, the generation, transfer and expression of chirality during the formation of hierarchical supramolecular structures are very complicated.

Figure. R. 7. a) CD (up) and UV-vis (down) spectra for L-1 (black line) and D-1 (red line) gels in DMSO/H₂O (v/v = 9/1), [L-1] = [D-1] = 2 mM, [BPEA] = 0.4 mM. b) UV-vis spectra for D-1/BPEA gel in DMSO/H₂O (red line) and 3.17×10⁻⁴ M BPEA in CHCl₃. 0.1 mm and 1mm cuvette were used in the UV-vis spectra measurement for gel and CHCl₃ solution respectively.

Here, we will give a detailed discussion on the origin of bisignate CD band in the 420 – 500 nm. In Figure R7a, the CD spectra for L-1/BPEA or D-1/BPEA gels in DMSO/H₂O showed very complicated exciton couplet. This situation is partly due to that the S₀-S₁ transition absorption with two vibronic peaks for BPEA molecule is close together. The split type CD signals for these two peaks have partial overlap. On the other hand, the strong CD signals for L-1 or D-1 molecules in the range of 200 – 420 nm will have interference in the CD signals of BPEA. In fact, the L-1/BPEA or D-1/BPEA formed semi-transparent gel in DMSO/H₂O, which will induce scattering effect in the CD measurement. This can be well illustrated in the Figure 4 of the manuscript, the CD value of the baseline was high. Considering the scattering effect in the CD measurement for the gel in DMSO/H₂O, we are very careful to analysis these split CD signals. In order to gain insight into the state of BPEA (monomer or

aggregate) in the gel, we gave a comparison of UV-vis spectra between D-1/BPEA gels in DMSO/H₂O and BPEA solution in CHCl₃ (Figure. R. 7b). The UV-vis spectrum of BPEA solution in CHCl₃ (good solvent for BPEA) showed two monomer absorption peaks at 438 and 464 nm. While the L-1/BPEA or D-1/BPEA showed two absorption peaks at 445 and 470 nm, which are bathochromic shift compared with that in CHCl₃. This indicated that BPEA molecules mainly existed in the form of J-like aggregation in the gel formed by L-1 or D-1 in DMSO/H₂O. The CD spectrum for L-1/BPEA showed two positive bisignate CD bands in the range of 420 – 500 nm. The first positive bisignate CD band: 470 nm (positive), 459 nm (negative), 464 nm (the crossover); the second positive bisignate CD band: 445 nm (positive), 429 nm (negative), 438 nm (the crossover). It should be noted that the crossover in this case is just located at the monomer absorption peak (438 nm), while the positive peak corresponds to the maximum absorption peak at 470 nm. This clearly indicated the positive exciton couplet, which is just opposite to the L-1 exciton, suggesting that the chirality transfer in a multi to multi-mode. Moreover, a mirror CD can be observed for the D-1/BPEA system. On the other hand, the second bisignate CD band is not well-resolved, mainly due to the interference from the strong CD signal of L-1 or D-1 in the range of 200 – 420 nm. It should be also noted that the peak seeming-like CD signal at about 490 nm arose from the scattering effect in the measurement process. The results strongly suggest that the supramolecular chirality rather than the chirality at the L-1 or D-1 is responsible for the ICD of BPEA.

7. “their CD spectrum” should be their CD spectra (pls check again other sl and pl as countable nouns)

Author reply: Thank you very much for your kind suggestion. In our revised manuscript, we have carefully checked the spelling for singular and plural and replaced the word “spectrum” with “spectra” when the noun is plural.

8. Two positive signals at 445 and 470 nm. I think, this is as a consequence of a bisignate CD band due to chirally assorted origin, so-called exciton couplet. I estimated the g_{abs} values at 445/470nm are ± 0.02 , respectively, for L-1 with BPEA. CPL at 500 nm along with weak vibronic CPLs at 520 nm is from the mixed gel 470 nm band. The CD sign at 470 nm is identical to that of CPL at 410 nm. Based on Stokes shift between the ground-state and photoinduced reorganization of emitters at the S1-state, if Kasha’s rule is applied. Actually, for L-1 alone gel CD sign at 380 nm is identical to that of CPL sign at 500 nm.

Author reply: Thank you very much for your comment. The g_{abs} values for 445 and 470 nm in L-1/BPEA are calculated as 0.0033 and 0.01 respectively according to the following equation:

$$g = \frac{\Delta A}{A} = \frac{\text{CD (mdeg)}}{32980A}$$

For L-1/BPEA, the CD signal at 470 nm is positive (not negative), so it is opposite (not same) to that of CPL at 410 nm. At the same time, for L-1 alone gel CD sign at

380 nm is opposite to that of CPL sign at 500 nm. The result is consistent with Kasha's rule.

9. It is unclear for me, where BPEA is placed to L-1/D-1 nano helix, whether it is inside or outside and whether isolated or assorted already. According to illustration of Scheme 1, BPEA is located into nano helix as isolated molecules, But the corresponding CD band of BPEA originates from the exciton couplet as I mentioned above, possibly due to chirally assorted BPEA molecules, possibly, not due to isolated BPEA.

Author reply: Thank you very much for your comment. Firstly, the X-ray diffraction and FT-IR measurements indicated that the addition of BPEA to L-1(or D-1) did not destroy the well-ordered packing of L-1 in the nanohelix. The fluorescent microscopy measurement further indicated that BPEA co-assembled with L-1 or D-1 in the nanohelix. If BPEA is located outside the nanohelix, the observable phase-separation will occur. So BPEA should be located inside the nanohelix. Secondly, in the CD spectra of L-1/BPEA and D-1/BPEA as shown in Figure 4b, the corresponding bisignate CD band of BPEA in the range of 440-500 nm originating from the exciton couplet indicated that the BPEA molecules should exist as some aggregates assorted in the nanohelix. On the other hand, the fluorescence spectra of L-1/BPEA or D-1/BPEA gel do not exhibit observable excimer emission of BPEA (see the Figure 3b and Figure S3b), which indicated that no serious aggregation of BPEA molecules in the nanohelix. So based on the CD and fluorescence spectra, we can conclude that small BPEA aggregates mainly existed in nanohelix. We have changed the scheme.1 in our revised manuscript.

10. g_{lum} at 410 nm $\pm 1.1 \times 10^{-2}$, g_{lum} at 510-520 nm $\mp 0.3 \times 10^{-2}$ (Fig5b, inset) but g_{lum} at 510-520 nm $\mp 3 \times 10^{-2}$ (Fig5c, plot). Pls confirm this inconsistency.

Author reply: Thank you for your comment. In the Figure 5c of our original manuscript, the unit for Y-axis is $g_{lum} \times 10^3$. In other words, $g_{lum} \times 10^3 = \mp 3$ at 510-520 nm, so the $g_{lum} = \mp 3 \times 10^{-3}$. In Figure 5b, the g_{lum} at 510-520 nm $\mp 0.3 \times 10^{-2}$. So the g_{lum} at Figure 5b is consistent with the g_{lum} at Figure 5c.

11. Pls disclose the corresponding PL signal simultaneously obtained the CPL spectra in Fig. 5.

Author reply: Thank you very much for your kind suggestion. In our revised manuscript, we have added the corresponding PL signal simultaneously obtained the CPL spectra. In every measurement, we adjusted the parameters of the apparatus to ensure the similar luminescence intensity.

12. In Fig S3b, PL spectrum of the BPEA film is strange for me. The normalized PL spectra revealed three major bands at 260, 610, and 640 nm. I assume the 640 nm band with a very narrow bandwidth of ≈ 10 nm is due to stray light when excited at 320 nm with a very narrow bandwidth of ≈ 5 nm due to ill-tuned PL instrumental origin. For example, the damaged grating or broad bandwidth for excitation, multiple scattering at the interface of air/film/substrate often cause this stray light. Should

re-measure PL using a specific cut-filter or moving excitation wavelength from 320 nm to several other wavelengths (300, 310, 330, 340nm) whether the 640 nm narrow PL band remains or retains or moves to blue or red associated with these exception wavelengths.

Author reply: Thank you very much for your kind suggestions. As you suggested, we have re-measured the PL spectra of BPEA film moving excitation wavelength from 320 nm to several other wavelengths including 300 nm, 310 nm, 340 nm, 400 nm and 500 nm in the absorption range of BPEA (shown in Figure R.8). In all the measurement process, the emission slit, excitation slit and PMT voltage are set as 5 nm, 5 nm, 700 V. When excited at 300 nm, 310 nm, 340 nm, the emission band at 640 nm moves to 600 nm, 620 nm, 680 nm, respectively. The corresponding wavelength of new emission band is twice wavelength of the excitation light. If moving excitation light to longer wavelengths such as 400 nm, 500 nm, the emission band at 640 nm will move to longer wavelength region out of the spectral range (500 – 750 nm). The experimental results indicated that the 640 nm band excited at 320 nm is due to stray light rather than to come from the emission of BPEA. In fact, we often come across the situation where emission band with twice wavelength of the excitation light exists. On the other hand, the wide emission band centered at 590 nm do not remove regardless of the excitation wavelength, which indicated that this emission band should come from the excimer emission of BPEA.

Figure. R. 8 (a) Fluorescent image of BPEA cast film and corresponding (b) UV-vis and (c) FL spectra. For the measurement of Fluorescent image of BPEA, the excitation light wavelength λ_{ex} is 325 ~ 375 nm. BPEA solution in $CHCl_3$ was dropped on the

surface of quartz plate and dried in air. For the measurement of emission spectrum of BPEA film, different excitation wavelength including 300 nm, 310 nm, 320 nm, 340 nm, 400 nm, 500 nm were used. The 600 nm, 620 nm, 640 nm, 680 nm emission peaks are attributed to the 1/2 fraction frequency fluorescence peaks. The emission slit, excitation slit and PMT voltage are 5 nm, 5 nm, 700 V.

13. Should disclose in Fig S3a, show excitation, dichroic, and long-pass filtered wavelengths for obtaining the image.

Author reply: Thank you very much for your suggestions. In the experiment, the excitation wavelength is in the range of 325 ~ 375 nm. Long-pass filter U-DAPL-LP was used in the experiment and the long-pass filtered wavelength is 425 nm. We did not use dichroic in the experiment.

14. In my computer, some messy codes and no fonts, possibly, symbol, nu and delta, are displayed. Should re-check manuscript.

Author reply: Thank you very much for your kind suggestion. In our revised manuscript, we have carefully re-checked the papers and revised the founded errors.

REVIEWERS' COMMENTS:

Reviewer #1 (Remarks to the Author):

I recognized appropriate efforts for revisions and answers by the authors. The revised version becomes acceptable.

Reviewer #2 (Remarks to the Author):

I appreciate the authors' effort to include most of the suggestions/comments made by the reviewers. In my opinion, the quality of the manuscript has been substantially improved and is now suitable for publication once below mentioned minor suggestions are considered.

1. Reviewer # 1, comment 1: I would like to mention that the possibility of energy transfer from the self-assembled donors to isolated acceptor monomer or aggregated monomer is depending on the concentration of acceptor and its aggregation properties. For more details, authors can refer Ref. 37.

2. I would like to point out that after modification also, Scheme 1 creates some confusion. According to the left panel of the Scheme, acceptor molecules are aggregated, whereas the zoomed portion (right panel) gives an impression that acceptor molecules are encapsulated in helix as isolated molecules. I request authors should address this issue.

Reviewer #3 (Remarks to the Author):

The authors try to revise the original manuscript as possible in line with three reviewers. I had a look at this revision and I think it is fully consistent with their claims. Although soft gel and xerogel materials hold a difficulty to fully characterize compared to homogeneous solution science, well-defined supramolecular science and solid-state science, the outcome of this paper is worth publishing.

Response to the comments of reviewers 2

Reviewer #2:

I appreciate the authors' effort to include most of the suggestions/comments made by the reviewers. In my opinion, the quality of the manuscript has been substantially improved and is now suitable for publication once below mentioned minor suggestions are considered.

1. Reviewer # 1, comment 1: I would like to mention that the possibility of energy transfer from the self-assembled donors to isolated acceptor monomer or aggregated monomer is depending on the concentration of acceptor and its aggregation properties. For more details, authors can refer Ref. 37.

Author reply: Thank you very much for your kind comments. We totally agree with your opinion that energy transfer from the self-assembled donors to isolated acceptor monomer or aggregated monomer is depending on the concentration of acceptor and its aggregation properties. We have carefully read the Ref. 37. In our present study, the concentration of BPEA acceptor is in the range of 0-0.4 mM and the solvent is DMSO/H₂O (v/v = 9/1). The spectral results indicated BPEA molecules existed mainly as small aggregates in nanohelix. In such situation, the energy transfer is through self-assembled donors to aggregates (multi to multi). If we increase the concentration of BPEA acceptor while keeping the donor concentration, the co-gel will collapse, which indicated that excess of BPEA may destroy the supramolecular assembly of L-1. So the concentration BPEA was just kept in the range of 0-0.4 mM for the study of chirality and energy transfer in the self-assembly system. Because BPEA molecules had no serious aggregation in such concentration range, their emission peak had no obvious or large red shift comparing to their isolated state.

2. I would like to point out that after modification also, Scheme 1 creates some confusion. According to the left panel of the Scheme, acceptor molecules are aggregated, whereas the zoomed portion (right panel) gives an impression that acceptor molecules are encapsulated in helix as isolated molecules. I request authors should address this issue.

Author reply: Thank you very much for your kind comments. The CD, UV-vis and FL spectra measurements indicated that the BPEA molecules existed mainly as aggregates assorted in the nanohelix. We have changed the existing form of BPEA acceptor molecules from isolated molecule to aggregates in the zoomed portion (right panel) of the Scheme.